# Mesoporous Carbon-Based Materials: A Review of Synthesis, Modification, and Applications

**Shahram Mehdipour-Ataei [1],\* and Elham Aram [2]**

1   Iran Polymer and Petrochemical Institute, Tehran P.O. Box 14965/115, Iran
2   Department of Polymer Engineering, Faculty of Engineering, Golestan University, Gorgan 49188-88369, Iran
\*   Correspondence: s.mehdipour@ippi.ac.ir

**Abstract:** Mesoporous carbon materials have attracted both academic and industrial interests because of their outstanding physical and chemical properties, such as high surface area, large pore-volume, good thermostability, improved mass transport, and diffusion. Mesoporous carbon materials with various pore sizes and pore structures can be synthesized via different methods. Their unique properties have made them a suitable choice for various applications, such as energy-storage batteries, supercapacitors, biosensors, fuel cells, adsorption/separation of various molecules, catalysts/catalyst support, enzyme immobilization, and drug delivery, in different fields. This review covers the fabrication techniques of mesoporous carbon structures and their typical applications in various fields and features a brief introduction of the functionalization and modification of mesoporous carbons.

**Keywords:** mesoporous materials; carbon; synthesis; application

## 1. Introduction

Porous carbon materials have been a focus of attention in chemistry, physics, and material sciences in recent decades because of their unique structures, physiochemical properties, superior chemical/mechanical stability, good electrical conductivity, and low cost. Generally, different carbon-based structures, including carbon nanotubes, fullerenes, porous carbons, have been widely applied for various applications, such as adsorbents for the purification of water [1,2]. Among them, porous carbons are attractive because their synthesis methods are very simple and the properties of these materials are much better than other carbon-based materials thanks to their porous structure. On the other hand, the well-porous structures support the incorporation of new functional agents such as organic or inorganic components within the porous channels or on the surface of the carbon walls, which significantly improve their properties and performance in various applications. Porous carbon materials have been applied for many years in various applications, including energy storage and conversion, gas separation, water purification, catalysis, and sensing [3–10]. According to the definition of the international federation of pure and applied chemistry (IUPAC), porous carbon materials can be divided into three groups according to pore size: microporous carbon (materials with pore diameter less than 2 nm), mesoporous carbon (materials with pore diameter between 2 and 50 nm), and macroporous carbon (materials with pore diameter larger than 50 nm) [11]. Among them, mesoporous carbon structures with a broad or narrow distribution of pores have recently attracted enormous attention in different fields because of high specific surface area, good chemical and mechanical properties, high thermal stability, large pore volume, and uniform and controllable porous structure. These outstanding properties enable mesoporous structures to be excellent candidates for various practical applications, including as a catalyst and its support, adsorption and separation, energy conversion and storage, environmental remediation, drug delivery, and biomedical applications [12–19]. At first,

mesopores in carbon structures were produced by enlarging micropores via oxidation during activation process, such as activated carbons, or at the interstices between carbon particles, such as carbon aerogels. The first technique led to a significant reduction in the yield of carbon, and the second one's morphology was difficult to control. Thus, new methods have been developed for producing of mesoporous carbons [20]. In this regard, the first study on the synthesis of mesoporous carbon materials was reported in 1999 by Ryoo et al. [21]. They used mesoporous silica as a hard template and sucrose as a carbon source to produce a mesoporous structure with high specific surface areas and tunable pore diameters between 2 and 50 nm. This invention has attracted much attention from many researchers for the preparation and potential application of mesoporous carbon materials. Rapid progress has been made in the field of mesoporous carbon material synthesis using the self-assembly of copolymer molecular arrays and carbon precursors, by Dai and Zhao' groups [22,23]. Different templates with various structures can be used for the preparation of mesoporous materials with a high level of control over their structures. For example, several mesoporous silica materials with various structures, such as SBA-1, SBA-15, SBA-16, MCM-48, KIT-6, and KIT-5, are applied as templates for the synthesis of mesoporous carbon materials with different mesoporous structures [24–26]. The production of mesoporous carbons through hard mesoporous silica templates has led to structures with surface areas of more than 2000 $m^2$ $g^{-1}$, a higher pore volume of up to 3.0 $cm^3$ $g^{-1}$, and tunable pore diameters, along with high conductivity. Furthermore, a similar hard-templating method was also used by Hyeon et al. for the fabrication of ordered porous carbon using phenolic resin as the source of carbon [27,28]. Numerous approaches are used to synthesize mesoporous carbon materials, including (a) activation methods (physical activation and chemical activation), (b) template methods (hard template and soft template), (c) carbonized method (the carbonization of polymer/polymer blends, organic aerogels, etc.), and (d) catalytic activation using metal ions. According to the preparation techniques and final structures, mesoporous carbon materials are classified into two types [29,30]: (a) ordered mesoporous carbon, synthesized in the presence of ordered mesoporous silica or triblock copolymer structure-directing species templates, and the resulting structure has uniform mesopores, which can be periodically arranged to show distinct X-ray diffraction lines, and (b) disordered mesoporous carbon, which can be produced by different preparation methods, such as thermal activation, catalytic activation using several metals, and the carbonization of mesoporous organic aerogels or polymer blends. The final structure has isolated or irregularly interconnected mesopores, and the distribution of pore sizes is relatively wider than ordered mesoporous carbons. Recently, the synthesis procedure for mesoporous carbons via magnesium oxide as a template has been industrialized and the carbons commercialized as magnesium oxide-templated mesoporous carbons [20]. This can lead to new applications in mesoporous materials. The different synthetic methods result in mesoporous carbon materials with broad or narrow pore-size distributions. They also introduce a lot of micropores into the structure. In addition, these techniques can produce diverse morphologies such as nanofibers, nanoparticles, nanosheets, and nanotubes in porous materials, which can meet all the needs of industrial applications [31–33]. It has been found that the performance of mesoporous materials in the various fields is affected mainly by structural properties such as the specific surface area, pore size, pore structure, and heteroatoms doping, such as nitrogen (N), phosphorus (P), sulfur (S), and boron (B) [34]. The pore size and pore structure can conduct the diffusion of various materials into the pores. For example, Sung et al. investigated the effect of the pore size of mesoporous carbons on the oxygen-reduction reaction performance, and their results revealed that the pore size had a significant effect on oxygen diffusion and the oxygen-reduction reaction activity [35]. The presence of mesopores in carbon structures can increase the contact area and thus facilitate the reaction of oxygen. Furthermore, the pore size and pore structure can manage the configuration of the various materials in the pores, thus affecting the effective surface area and ultimately affecting the performance of materials [36]. For instance, some researchers have loaded the protease enzyme on mesoporous carbons

with different pore sizes for the oxidation of glucose. They indicated that the pore size of the mesoporous carbons had a significant effect on the diffusion of the enzyme into the pores and the immobilization and configuration of enzyme in the mesopores, thereby influencing the performance of carbons. The high porosity in carbon structures brings an enlarged specific surface area, which increases the accessibility of various reactants to the surface of mesoporous carbons [37]. Furthermore, the physical and chemical properties and the efficiency of mesoporous carbon structures can be controlled by the functionalization of the porous materials with organic or inorganic groups. For example, the introduction of heteroatoms, including B, N, S, or P, in the carbon structure can change the specific surface area, the electronic properties, and the efficiency of the materials, which are critical for various applications. Therefore, mesoporous carbon structures with or without different functional groups offer unique properties over other porous materials [38–40]. In general, the technology and process for the synthesis of mesoporous carbons is a challenging task and encompasses the effective utilization of the multidisciplinary scientific fields, including polymer, materials, chemistry, physics, and chemical and mechanical engineering.

In this review, the general techniques for the preparation of mesoporous carbon structures are described. Then, different modification methods for mesoporous carbon materials are reviewed. In addition, various applications of these materials, including as catalysts, as adsorbents, for enzyme immobilization, for drug delivery, as supercapacitors, as Li-batteries, as sensors, as fuel cells, and so on, according to studies in the literature in recent years, are discussed. Following these, the research challenges and future directions on the development of mesoporous carbon materials for different applications are proposed.

## 2. Synthesis of Mesoporous Carbons

Because of the various applications of mesoporous carbon materials, several methods have been developed for the synthesis of mesoporous carbon over the past few decades. Depending on the pore size and its distribution and on the final structure of mesoporous material, several approaches can be used for mesoporous carbon preparation, which are described in the following section.

### 2.1. Activation Methods

Activation techniques, which can be classified into chemical and physical methods, are the most used approaches to prepare mesoporous carbon (Figure 1). The physical activation process includes the thermal pyrolysis of materials at a relatively low temperature (typically 400 to 900 °C) under inert atmosphere (nitrogen or helium) and then the controlled gasification of the obtained chars from carbonization at a higher temperature (800 1000 °C) with oxidizing agents such as water (steam), carbon dioxide, air, or any mixture of these gases to produce the porous structures [41,42]. Unlike commonly applied physical activation, the carbonization and activation in chemical activation are accomplished in one-step process. A typical chemical activation consists of the thermal decomposition of carbon precursor impregnated with suitable chemical agents at temperatures ranging from 400 °C to 700 °C [43]. The most widely used industrial activating agents in the chemical activation process are zinc chloride ($ZnCl_2$), aluminum chloride ($AlCl_3$), magnesium chloride ($MgCl_2$), sodium hydroxide (NaOH), potassium hydroxide (KOH), sodium carbonate ($Na_2CO_3$), and phosphoric acid ($H_3PO_4$) [43,44]. The type of chemical agent is one of the important and effective parameters of the mechanism and temperature of the activation process. The advantages of physical activation include the greenness of the process, availability of activating agents, and ease of technology. The chemical activation method has various advantages, such as relatively low activation temperature, higher carbon yield, low energy cost, shorter activation time, and the production of mesoporous carbons with a superhigh specific surface area in comparison to the physical activation process [45–47]. Generally, in the practical applications, physical and chemical activation procedures are combined to obtain better activation effects.

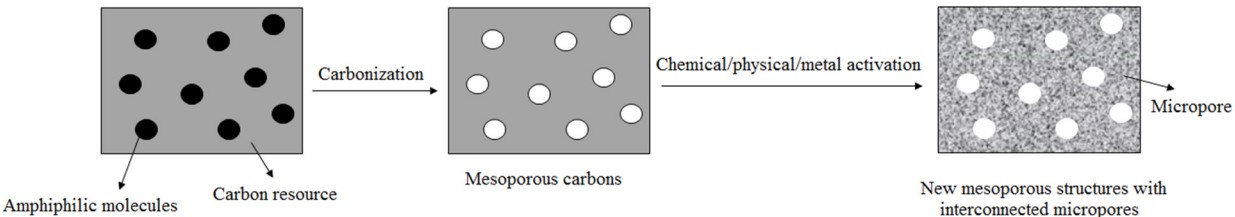

**Figure 1.** Schematic diagram of activation method for synthesis of porous carbon.

## *2.2. Catalytic Activation Using Metal Ions*

In this technique, the activation process of carbon precursors is carried out in the presence of metal ions such as Fe, Ni, Co, and so on as catalysts (Figure 1). These catalysts play an important role in the activation rate and in controlling of porous structure in carbon materials. Generally, catalysts accelerate the activation process of the chars and lead to the production of a structure with a larger volume of mesopores [48]. However, when such mesoporous carbons are actually used in an aqueous solution, the elution of metal cation into the solution might occur, which could be a serious problem under some circumstances.

## *2.3. Template Methods*

One of the most important and widely used methods for the synthesis of mesoporous carbon with well-controlled mesopores is the templating technique. Two well-established templating approaches, hard- and soft-templating methods, are applied for the preparation of mesoporous carbon (Figure 2). In the soft-templating method, amphiphilic molecules (surfactant molecules or block copolymers, metal organic frameworks), which act as the templates, are first self-assembled with carbon precursor molecules, through hydrogen bonds or other interactions. Subsequently, the amphiphilic molecules can be removed through the calcination process at high temperatures to generate mesopore structures with high controllability in the pore structures and dimensions [49]. In this approach, templates act mainly as molds for the replication of mesoporous carbon materials. The structure of the obtained pores is determined by various parameters, and the most important of them include the molecular properties of the tempering agents, reactant concentrations, solvents, and temperature. The hard-templating method usually uses pre-synthesized organic or inorganic templates such as magnesium oxide, mesoporous silica, and colloidal silica, and the carbon precursors are infiltrated into the pores of the templates. After carbonization under appropriate temperature and template removal through the chemical etching under acid or an alkaline solution, the mesoporous carbon with a large volume and surface area of pores is produced. Compared with the soft-templating approach, the hard-templating approach is more tedious, has a relatively long synthesis time, has greater difficulties in regulating pore size, and has a high production cost [50].

## *2.4. Carbonized Method*

Another method for the preparation of mesoporous carbon materials is carbonization. Carbonization is the conversion of carbon-containing materials into carbonization product under high temperature. Recently, the carbonization of polymer blends composed of a carbonizable polymer and a pyrolyzable polymer has become of interest for the synthesis of porous carbon. A polymer blend is a mixture of two or more types of polymers. When a polymer blend that consists of two types of polymers with different degrees of thermal stability is carbonized, the thermally unstable polymer (pyrolyzable polymer) will be completely eliminated in pyrolysis to leave pores in the carbon matrix of the thermally stable polymer (carbonized polymer) (Figure 3). The size and volume of pores in the final porous carbon can be controlled with the blending ratio of the component polymers [48].

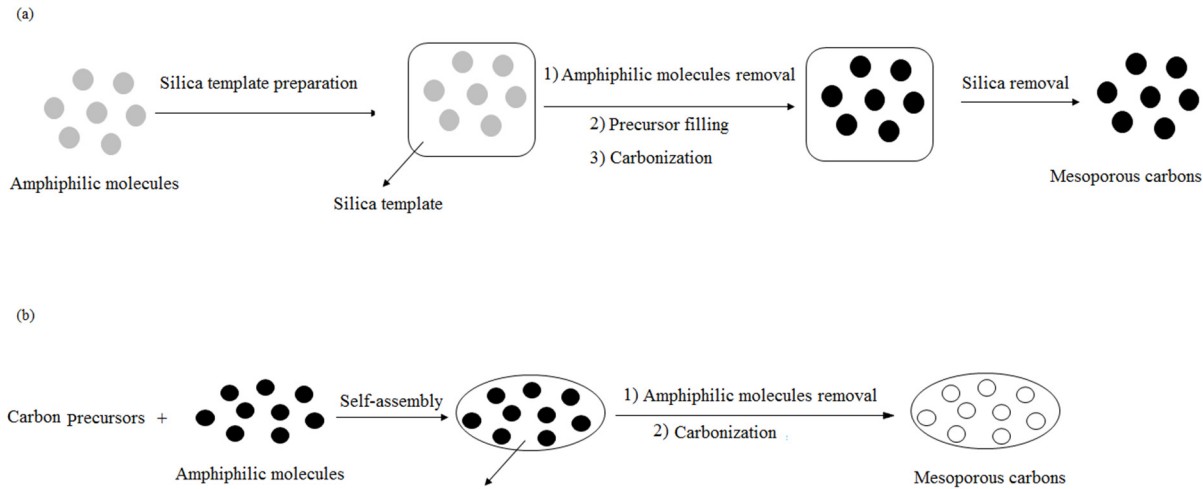

**Figure 2.** Scheme of the hard-templating method (**a**) and soft-templating method (**b**) in the synthesis of mesoporous carbon materials.

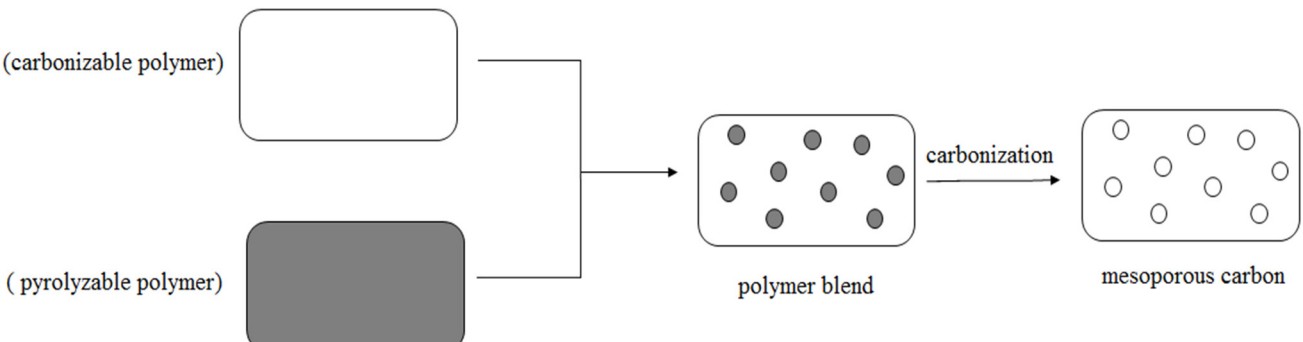

**Figure 3.** Schematic diagram of polymer blend carbonization.

Cyrogels are another category of materials that can be used for the preparation of mesoporous carbons. Carbonization of the freeze-dried cyrogels of polymer blend with various contents can produce porous structures with broad pore-size distributions [51]. The carbonization of organic aerogel materials prepared by sol-gel reaction is one more method for the synthesis of mesoporous carbons with high surface areas. Organic aerogels are produced by the polymerization of organic monomers such as resorcinol, melamine, formaldehyde, isocyanates, and so on. The uniform mesopores can be formed, for example, during the pyrolysis of resorcinol and formaldehyde [52]. The porosity and surface area of the final products can be controlled by changing the synthetic parameters, such as the reactant concentration ratio or the temperature.

## 3. Modification of Mesoporous Carbon Materials

The mesoporous carbon materials have excellent physicochemical properties with tunable pore diameter, large surface area, and high porosity. However, the hydrophobic nature and inertness of mesoporous carbons might not be suitable for various applications, including separation, adsorption, catalysis, and electronics. Therefore, the functionalization of the carbon materials is crucial to changing the hydrophobic/hydrophilic properties of the carbon materials and developing their potential applications. The introduction of functional groups into carbon materials can be obtained in situ during synthesis, or ex situ during postsynthesis reaction. The first method can produce structures with a relatively uniform distribution of the functional groups and low structural order. In this technique, functionalization can be conducted by using heteroatom-containing monomers in the sol-gel synthesis of either soft-templated carbons or carbon gels. The second one shows a high variability in

the incorporation of functional groups [50]. Many efforts have been devoted to the surface modification of porous carbon materials in recent years. A series of functional groups can be introduced onto the carbon material surface through surface oxidation and/or activation, halogenations, sulfonation, grafting through diazonium chemistry, and such [53–55]. Among them, surface oxidation is one of simplest techniques for modifying the mesoporous carbon surface, through which oxygen-containing groups such as carboxylic acids, esters, and quinones are generated. Oxidation can be conducted in different media, such as the liquid phase (nitric acid, hydrogen peroxide, ammonium persulphate) or in the gas phase (oxygen, ozone, nitrogen oxides) [56]. Gas-phase treatments are more effective for oxygen incorporation and also increase the micropore and mesopore volumes, as well as the micropore average size in comparison to the liquid medium [56]. Vinu et al. analyzed the ability of lysozyme adsorption using mesoporous carbon and oxidized mesoporous carbons with ammonium persulfate solution [57]. Their results showed that the carboxyl group functionalization onto ammonium persulfate-treated mesoporous carbon provided higher adsorption ability because the adsorbent molecules easily access the interior portion of mesoporous carbon materials. The presence of carboxyl groups may hinder the release of protein biomolecules from porous channels of carboxyl functionalized mesoporous carbon, which results in improved protein adsorption. The surface chemical nature of mesoporous carbon materials can be changed by the incorporation of heteroatoms such as oxygen, nitrogen, phosphorus, sulfur, boron, etc. by using heteroatom-containing precursors, which leads to improved properties for carbon materials. Oxygen-doped mesoporous carbon materials show an improvement in the hydrophilicity of the surface, which can act as active sites in catalytic reactions or in the selective adsorption of cationic materials and also facilitate the dispersion of metal [58,59]. Nitrogen doping in carbon materials produces a basic property on its surface. It can improve the anion-exchange properties, increase the adsorption of acidic species, and enhance the catalytic activity of mesoporous carbons by means of redox reactions. Phosphorus-doped mesoporous carbon materials reveal an increase in oxidation resistance and improve the supercapacitor energy density [50,60].

## 4. Applications of Mesoporous Carbon Materials

Mesoporous carbon materials are of major interest in many applications, such as separation, catalysis, and energy storage, because of their excellent physicochemical properties and high surface area. In the following sections, an accessible summary of the most important applications of mesoporous carbons is collected in Table 1, followed by some related explanations.

**Table 1.** Summary of the various applications of mesoporous carbon.

| Materials | Synthetic Method | Application | Purpose | Reference |
|---|---|---|---|---|
| Mesoporous carbon nanoparticles with polyacrylic acid capping | Soft template | Drug delivery | Improvement the release of doxorubicin into HeLa cells | [21] |
| Nitrogen-doped mesoporous carbon | Carbonization and chemical activation | Adsorbent | Carbon dioxide adsorption in fixed bed | [61] |
| Magnetic mesoporous carbon | Chemical activation and sol-gel method | Adsorbent | Remove of methyl blue and methyl orange from wastewater | [62] |
| Thiol-functionalized mesoporous carbons | Hard template | Adsorbent | Adsorption of bivalent heavy-metal ions (Cd, Cu, Pb, Zn) from the aqueous solutions | [63] |
| Mesoporous carbons | Activation | Adsorbent | Elimination of sulfur dioxide from flue gases and natural gases | [64] |

**Table 1.** *Cont.*

| Materials | Synthetic Method | Application | Purpose | Reference |
|---|---|---|---|---|
| Amine-functionalized mesoporous carbon | Carbonization | Adsorbent | Adsorption of carbon dioxide from flue gas | [65] |
| Magnetic ordered mesoporous carbon | Soft template | Adsorbent | Adsorption of sulfur from oil | [66] |
| Mesoporous carbon | Pyrolysis | Adsorbent | Adsorption of carbon dioxide | [67] |
| Ordered mesoporous carbon | Soft template | Adsorbent | Desulfurization of gasoline | [68] |
| Mesoporous carbon | Carbonization | Adsorbent | Carbon dioxide/methane separation | [69] |
| Nickel ferrite bearing nitrogen-doped mesoporous carbon | Carbonization | Adsorbent | Remove of mercury (II) from aqueous solution | [70] |
| Nitrogen- and phosphorus-codoped ordered mesoporous carbon | Hard template | Supercapacitor | Supercapacitor electrode | [71–73] |
| Mesoporous carbon decorated graphene | Template | Supercapacitor | Supercapacitor electrode | [74] |
| Ruthenium oxide-iron oxide nanoparticles embedded ordered mesoporous carbon | Template | Supercapacitor | Supercapacitor electrode | [75] |
| Mesoporous carbon aerogels | Ionothermal carbonization | Supercapacitor | Supercapacitor electrode | [76] |
| Mesoporous carbon | Soft template | Supercapacitor | Supercapacitor electrode | [77] |
| Mesoporous carbon | Hard template | Supercapacitor | Electrodes for solid-state supercapacitors | [78] |
| Mesoporous carbon | Carbonization of cellulose aerogel | Supercapacitor | Supercapacitor electrode | [79] |
| Mesoporous carbon | Carbonization and activation | Electrochemical double-layer capacitors | Electrode materials for electric double-layer capacitor (EDLC) supercapacitors | [80] |
| Mesoporous carbon | Carbonization | Supercapacitor | Supercapacitor electrode | [81] |
| Mesoporous carbon/manganese dioxide nanocomposite | Hard template | Supercapacitor | Supercapacitor electrode | [82] |
| Microporous and mesoporous carbon | Hard template | Lithium-sulfur batteries | Cathode electrode | [83] |
| Ordered mesoporous carbon fiber sulfur composite | Carbonization | Lithium-sulfur batteries | Cathode electrode | [84] |
| Mesoporous carbons with metal sulfide and metal oxide | Hydrothermal | Lithium-ion batteries | Anode for lithium-ion batteries | [85] |
| Iron oxide nanoparticles with mesoporous carbon and nitrogen-doped carbon | Carbonization | Lithium-ion battery | Anode for lithium-ion batteries | [86] |
| Mesoporous carbon nanofibers | Carbonization | Lithium-ion battery | Anode of lithium-ion batteries | [87] |
| Mesoporous carbon/zinc oxide | Carbonization | Lithium-ion battery | Anode of lithium-ion batteries | [88] |
| Mesoporous carbon/Tin oxide nanoparticles | Hydrothermal | Lithium-ion battery | Anode of lithium-ion batteries | [89] |
| Graphitized mesoporous carbon | Hard template | Lithium-ion battery | Anode of lithium-ion batteries | [90] |
| Foamed mesoporous carbon/silicon composite | Foaming and carbonization | Lithium-ion battery | Anode of lithium-ion batteries | [91] |

**Table 1.** *Cont.*

| Materials | Synthetic Method | Application | Purpose | Reference |
|---|---|---|---|---|
| Ordered mesoporous carbons | Hard template | Fuel cell | Supports of electrocatalysts for direct methanol fuel cells | [92–94] |
| Iron-doped mesoporous carbon | Template method | Fuel cell | Catalyst for oxygen-reduction reaction | [95] |
| Transition metal (iron, cobalt, manganese) and nitrogen-doped mesoporous carbons | Pyrolysis | Fuel cell | Cathode catalysts for anion-exchange membrane fuel cells | [96] |
| Nitrogen- and sulfur-dual-heteroatom-doped ordered mesoporous carbon | Template | Fuel cell | Electrocatalyst for oxygen-reduction reaction | [97] |
| Mesoporous carbon | Carbonization | Fuel cell | Platinum support as anode and cathode catalyst | [98,99] |
| Mesoporous carbon | Hard template | Methanol fuel cell | Anodic catalyst | [100] |
| Mesoporous carbon nitride | Hard template | Fuel cell | Electrode modification for the oxygen-reduction reaction | [101] |
| Microporous/mesoporous carbon | Carbonization and activation | Microbial fuel cells | Anode material | [102] |
| Mesoporous carbon | Carbonization | Proton exchange membrane fuel cell | Cathode catalyst support for oxygen-reduction reaction | [103] |
| Platinum/palladium/mesoporous carbon | Carbonization | Biosensor | Non-enzymatic amperometric sensing of glucose | [104,105] |
| Nitrogen-doped porous carbon nanosheets | Pyrolysis | Biosensor | Electrocatalyst for reduction of hydrogen peroxide | [106] |
| Ordered mesoporous carbon | Hard template | Electrochemical sensors | Modification of glassy carbon electrode for detection of cysteine | [107] |
| Mesoporous carbon/zirconium-based metal organic frameworks/laccase | Carbonization | Biosensor | Modifier electrode for tetracycline detection | [108] |
| Quantum dots/ordered mesoporous carbon | Carbonization | Electrochemi-luminescence biosensor | Detection of hydrogen peroxide in live cells | [109] |
| Ordered mesoporous carbons | Hard template | Biosensor | Modification of glassy carbon for determination of dopamine | [110] |
| Mesoporous carbons | Hard template | Electrochemical sensor | Modification of glassy carbon electrode for detection of dopamine | [111] |
| Boron- and nitrogen-doped mesoporous carbons | Carbonization | Electrochemical sensor | Electrode modifier to detect isoniazid | [112] |
| Mesoporous carbon nitride | Hard template | Catalyst | Catalytic application as solid bases | [113,114] |
| Sulfur, phosphorus, boron and iron-doped mesoporous carbon | Carbonization | Catalyst | Oxygen-reduction catalyst in microbial fuel cells | [115] |
| Mesoporous/macroporous carbon | Hard template | Catalyst | Catalyst support for the conversion of cellulose to polyols | [116] |
| Ferrum- and nitrogen-doped mesoporous carbon | Soft template | Catalyst | Catalyst for oxygen reduction in fuel cells | [117] |
| Mesoporous carbon | Carbonization | Catalyst | Support of platinum nanoparticles for the oxidation of organic fuels | [118] |
| Mesoporous carbon | Hard template | Catalyst | Support of Fe catalyst for $CO_2$ hydrogenation to liquid hydrocarbons | [119] |

**Table 1.** *Cont.*

| Materials | Synthetic Method | Application | Purpose | Reference |
|---|---|---|---|---|
| Magnetic mesoporous carbon | Thermal carbonization | Catalyst | Heterogeneous Fenton catalyst | [120] |
| Nitrogen- and sulfur-doped mesoporous carbon | Hard template | Catalyst | Cathode catalysts for direct biorenewable alcohol fuel cells | [121] |
| Fluorine-doped mesoporous carbon | Sol-gel | Catalyst | Oxygen-reduction catalyst | [122] |
| Mesoporous carbon/Ferro ferric oxide | Carbonization | Enzyme immobilization in biosensor | Modification of screen-printed carbon electrodes to detect organophosphorus pesticides | [123] |
| Mesoporous carbon nitride | Hard template | Enzyme immobilization in biosensor | Detection of phenol and catechol in compost bioremediation | [124] |
| Chitosan-entrapped mesoporous carbon nanocomposite | Template | Enzyme immobilization in biosensor | Modifier of electrode for enhanced amperometric sensing of glucose sensor | [125] |
| Mesoporous carbon | Organic-organic self-assembly | Enzyme immobilization in sensors | Support for redox protein immobilization | [126] |
| Ordered mesoporous carbon | Carbonization | Enzyme immobilization | Electrodes for biosensing | [127] |
| Mesoporous Carbon | Carbonization | Enzyme immobilization | Modifier of carbon screen-printed electrode and support for enzyme immobilization | [128] |
| Mesoporous carbon | Hard template | Enzyme immobilization in sensors | Modification of glassy carbon electrode for glucose biosensor | [129] |
| Magnetic mesoporous carbon | Hard template | Enzyme immobilization | Support for immobilization of laccase in phenol and p-chlorophenol removal | [130] |
| Mesoporous carbon | Carbonization | Enzyme immobilization | Fabrication of glucose-sensing electrodes | [131] |
| Magnetic mesoporous carbon | Carbonization | Drug delivery | Delivery of doxorubicin into cancer cells and improvement of cancer cell killing ability | [132–134] |
| Mesoporous carbon with polyethylenimine and folic acid | Hydrothermal | Drug delivery | Improvement the uptake of nanoparticles by HeLa cells | [135] |
| Magnetic mesoporous carbon | Carbonization | Drug delivery | Enhancement of tumor cell ablation capability | [136] |
| Phospholipid coated ordered mesoporous carbon | Carbonization | Drug delivery | Delivery of doxorubicin into MCF-7 cells | [137] |
| Nitrogen-doped mesoporous carbon nanoparticles | Soft template | Drug delivery | Improvement the adsorption capacity of hydroxycamptothecin | [138] |
| Polyethylene glycol-functionalized oxidized mesoporous carbon | Hydrothermal | Drug delivery | Inhibition the growth of cancer cells and improvement the therapeutic efficiency | [139] |
| Hyaluronic acid modified mesoporous carbon nanoparticles | Carbonization | Drug delivery | Delivery of doxorubicin and verapamil and improvement the therapeutic effect on HCT-116 tumor in BALB/c nude mice | [140] |
| Arginine-glycine-aspartic acid peptide-conjugated mesoporous carbon | Carbonization | Drug delivery | High doxorubicin loading and improvement the therapeutic efficacy toward PC3 cells | [141] |

### 4.1. Adsorption

Nowadays, the carbon materials with good physicochemical properties, well-ordered mesopores, high pore volume, and high surface area provide crucial applications in the field of the adsorption and separation of gases molecules, biomolecules, and aqueous pollutants. Most pristine mesoporous carbon materials have a low adsorption capacity. Therefore, the functionalization of the carbon materials is required to change the hydrophobic and hydrophilic properties of the carbon surface and improve the adsorption capacity for certain molecules. In this regard, Yaumi et al. [61] prepared mesoporous nitrogen-doped carbon through carbonization and chemical activation with coconut shell, glucosamine, and KOH as the activating agent for $CO_2$ adsorption in a fixed bed. They reported that coconut shell (CS) was as a precursor, whereas glucosamine (GA) was as the nitrogen source. According to their results, the adsorbent showed maximum adsorption capacity of 4.23 mmol/g at 30 °C and 1 bar. They indicated that $CO_2$ adsorption capacity of mesoporous carbon can be improved by incorporating nitrogen functionalities into the carbon structure through glucosamine. In another study, magnetic mesoporous activated carbon was synthesized, and its application for the removal of methyl blue and methyl orange from dye solutions was investigated by Azam et al. [62]. They reported that activated carbon was prepared from rice husk by $ZnCl_2$ chemical activation and the mesoporous structure introduced by synthesizing KIT-6 template [62]. They used iron nitrate (III) as a precursor for imparting magnetic behavior. They also compared the adsorption efficiency of magnetic mesoporous activated carbon with mesoporous activated carbon and activated carbon by using all of them for the adsorption of methyl blue and methyl orange solutions. Their results showed that the magnetic mesoporous activated carbon had more adsorption capacity than other prepared adsorbents because of its functionalization by magnetite, which created more active sites and improved adsorption performance (Table 2). Furthermore, they indicated that the adsorption process was more efficient in an acidic medium than in a neutral or basic medium.

**Table 2.** Adsorption of methyl blue by the various adsorbents.

| Adsorbents | Adsorption (%) |
|---|---|
| Magnetic mesoporous activated carbon | 98.5 |
| Mesoporous activated carbon | 66.9 |
| Activated carbon | 76 |

Michalak-Zwierz et al. [63] prepared mesoporous carbons and studied the effect of the functionalization of mesoporous carbon materials with thiol-contained polyhedral oligomeric silsesquioxane (POSS) on the adsorption of heavy-metal ions such as Cu, Zn, Pb, and Cd in water. In their work, ordered mesoporous carbon were obtained by hard templating using SBA-15 and subsequent oxidization by ammonium persulphate. Then, the surface modification of mesoporous carbons was completed by octa(3-mercaptopropyl) polyhedral oligomeric silsesquioxane to decorate the surface with thiol groups. They reported that thiol functionalization did not change the adsorption efficiency of Cu(II) and Pb(II) as compared with pristine mesoporous carbon materials. However, the amount of Cd(II) absorption by thiol-functionalized mesoporous carbon was about four times less than pristine mesoporous carbon, whereas the amount of Zn(II) absorbed by thiol-functionalized mesoporous carbon was two times more than mesoporous carbons (Figure 4). In addition, they showed that the amounts of metal adsorbed (except Cd) in the range of 0–15 mg/L were higher for thiol-functionalized mesoporous carbon as compared with pristine mesoporous carbon. This was attributed to the strong chemical interaction between thiol groups and metal ions.

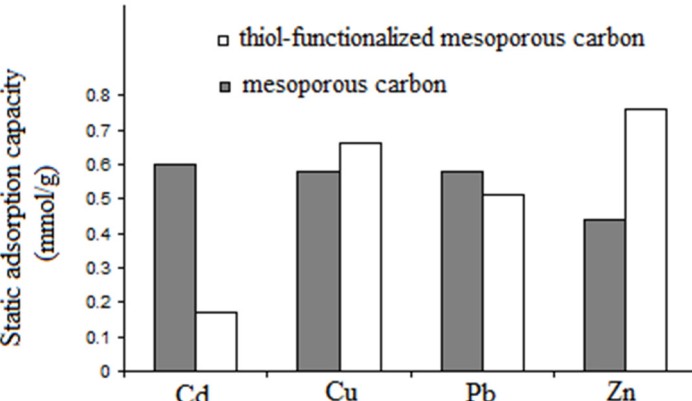

**Figure 4.** Static adsorption capacity of mesoporous carbon and thiol-functionalized mesoporous carbon.

In another work, the researchers prepared a new adsorbent for the elimination of $SO_2$ from flue gases and natural gases based on oil-tea shells-derived mesoporous carbons [64]. The obtained porous carbons had high specific surface areas (1195–1449 $m^2$ $g^{-1}$), mesoporous with pore sizes centered at 4–6 nm, and good $SO_2$-adsorption properties. They indicated that their carbon adsorbents showed a high $SO_2$-adsorption capacity of 10.7 mmol/g at 298 K and 1.0 bar, with an outstanding $SO_2/CO_2$, $SO_2/CH_4$, and $SO_2/N_2$ selectivity of 32, 127, and 2349, which was more than twice the absorption of $SO_2$ by commercial ordered mesoporous carbon CKM-3 (5.1 mmol/g). According to their density functional theory calculations, the presence of $-NO_x$ and $-OH$ groups in the carbon structure could form a strong interaction with $SO_2$ molecules and thus improve the $SO_2$-adsorption capacity and selectivity. In order to obtain the high-efficiency capture of $CO_2$, various excellent amine-based functionalized adsorbents have been developed. In this case, Zhao et al. synthesized mesoporous carbon with the large pore volume of 3.1 $cm^3$/g by direct carbonization of the mixture of adipic acid and zinc powder without further chemical/physical activation and then modification with tetraethylenepentamine to prepare solid amine adsorbent [65]. The results showed that when the mesoporous carbon support was modified by 80 wt% tetraethylenepentamine, the adsorbent had a high $CO_2$ capture amount of 5.3 mmol/g at 75 °C with excellent cycling stability. Apart from these, several other reports on the adsorption of different molecules have been seen in the literature [66–70]. As discussed above in this review, the mesoporous carbon materials with high specific areas are among the best materials to use as adsorbents. Furthermore, the surface modification of mesoporous carbons increases adsorption capacity. Therefore, the future challenge is to synthesize highly pure functionalized mesoporous carbons without using expensive reagents.

### 4.2. Supercapacitors

Supercapacitors are energy-storage devices that provide a high power density (410 kW/kg) and a fast charge/discharge process with long cycling life. A supercapacitor consists of two electrodes separated by a porous matrix (separator). In general, two types of supercapacitors are developed, including a pseudocapacitor and a double-layer electrical capacitor (EDLC). A pseudocapacitor is a type of supercapacitor that uses metal oxide or conducting polymer electrodes and stores electrical energy through faradaic reactions, whereas a double-layer electrical capacitor uses carbon electrodes and stores electrical energy by intercalating charges at the electrode–electrolyte interface, forming the double layer of charges [71]. In most cases, EDLCs have higher power density and charge/discharge cycling performance than pseudocapacitors, whereas their energy-storage capacity is at a lower level than that in pseudocapacitors [71]. The usual materials used in capacitor electrodes should have a large surface area to improve the electrochemical performance of supercapacitors. Recently, mesoporous carbons garnered much attention as

electrodes of supercapacitors because of their ease of operation, high chemical stability, large pore sizes, and high specific surface areas [72]. In this regard, many studies have been conducted [73–82], and descriptions of selected ones are the subject of the following section: Lufrano et al. [78] prepared mesoporous CMK-3-type carbon using SBA-15 silica as a template material and sucrose as the carbon source and used them as electrodes for solid-state supercapacitors. The results of this experiment showed that the use of CMK-3 mesoporous carbon could improve specific capacitance by approximately 68% when compared with an activated carbon (Norit A Supra Eur) that used as a reference material. Furthermore, the performance of a double-layer capacitance of the mesoporous carbon was higher than that of the reference carbon (Table 3).

**Table 3.** The specific capacitance and performance of mesoporous carbons and activated carbon.

| Sample | Specific Capacitance (F g$^{-1}$) | Performance (µF cm$^{-2}$) |
|---|---|---|
| Mesoporous carbon | 132 | 12.05 |
| Activated carbon | 80.5 | 5.3 |

This behavior could be attributed to high pseudocapacitance and improved properties, such as the pore structure, accessibility of pores to the electrolyte, and surface wettability by the surface functional groups of mesoporous carbons as compared with activated carbon. Similarly, Tian et al. [79] prepared porous carbons with mesopore/micropore hybrid structure-based electrodes for supercapacitors in the presence of cellulose as a carbon source through carbonizing cellulose aerogel at a temperature of 700 °C without the addition of an activating reagent. The produced porous carbons had a surface area of 646 m$^2$ g$^{-1}$ and a pore volume of 0.4403 m$^3$ g$^{-1}$, with a pore structure that consists of the micropores with a diameter of 1.49 nm and mesopores with pore diameters in the range of 2.25–3.32 nm. The results of electrochemical tests revealed that the porous carbon electrodes had good cycling stability and a comparably high electrochemical performance with a large specific capacitance (195 F g$^{-1}$) at a discharge current density of 0.1 A g$^{-1}$. They indicated that the mesoporous structure can provide channels for the easier penetration of electrolyte ions into the pores, while the microporous structure can store more electrolyte ions. In another study, by Li and his coworkers [80], mesoporous carbons by cylinder and gyroid pore structures had been synthesized and used as electrode materials for supercapacitors. For this purpose, resol-phenolic resin was used as the carbon precursor and poly(ethylene oxide-block-caprolactone) was used as the template, with a specific weight ratio of resol to PEO-PCL (60:40 for hexagonal cylinder structures; 55:45 for gyroid structures (Figure 5). They found that the mesoporous carbon with gyroid structure had specific capacitances higher than those of the cylinder carbon samples, which could be related to the interconnected mesopores of the gyroid-type activated mesoporous carbons that create a higher effective adsorption surface area.

**Figure 5.** Preparation of mesoporous carbons.

Mesoporous carbon nanofibers had been prepared for supercapacitor application by Dong et al. [81] through the electrospinning with phase separation and through carboniza-

tion technology. For this purpose, the researchers used polyacrylonitrile as the carbon precursor and the copolymer of polyacrylic acid-b-polyacrylonitrile-b-polyacrylic acid as pore forming additive in the spinning solution, and then nanofibers were carbonized at 800 °C to fabricate mesoporous carbon nanofibers. They investigated the effect of block copolymer content on electrochemical performance. According to their results, mesoporous carbons showed a maximum specific capacitance of 256.3 F g$^{-1}$ when the block copolymer content was 0.6 g, because of its large surface area and mesoporous structure, which could supply more active sites for ion adsorption to improve electrochemical capacitance. They revealed that a supercapacitor assembled with mesoporous carbon nanofibers could maintain 90% of capacitance after 80,000 cycles. In order to improve the capacitive performance of mesoporous carbons as electrodes in capacitors, their composites with metal oxides nanostructures have been offered. In this case, Mohammadi et al. prepared a novel nanocomposite based on mesoporous carbon and $MnO_2$ for the electrode material of supercapacitor [82]. Their results showed that $MnO_2$ uniformly distributed on the surface of mesoporous carbon, giving a porous structure with specific area of 185 m$^2$ g$^{-1}$. Mesoporous carbon/$MnO_2$ nanocomposite showed a considerable specific capacitance of 292 F g$^{-1}$ at a current density of 0.5 A g$^{-1}$, compared with mesoporous carbon. Furthermore, capacitors with a mesoporous carbon/$MnO_2$ nanocomposite electrode exhibited good rate performance and notable cycle durability. Although the mesoporous carbons offer good performance in the supercapacitors, it is still a challenge to obtain a cost-effective way for the large-scale production of cheap electrode materials that is both highly stable and reliable. Therefore, the future task is to produce pure mesoporous carbons without any contamination in a more economical way but without using harsh conditions and expensive reagents.

### 4.3. Lithium-Ion Batteries

Rechargeable batteries, especially lithium-ion batteries, have been widely used in portable electronics and electric vehicles in the past decades. Electrodes are made of materials that are conductors, and carbon is one of the suggested conductor materials. One of the problems of using carbon in electrodes is that the Li-ion penetration is difficult. Therefore, the main challenge of lithium batteries is related to the slow solid-state diffusion. In order to solve this problem, the use of porous materials is suggested, which can improve charge transfer and mass transport processes. Mesoporous materials with a 3D architecture are suitable as active materials or conductive agents in both anodes and cathodes of lithium-ion batteries. In this case, numerous studies are done [83–91]. For example, Shi et al. prepared mesoporous carbonized nanofibers via the electrospinning technique and thermal treatment, using lignin as the carbon source, polyacrylonitrile to improve the "spinnability" of the solution, and triblock copolymer Pluronic P 123 as the template [87]. They used mesoporous carbons as anodes of lithium-ion batteries and found that the mesoporous materials provide fast transport channels and reduce the diffusion distance of electrolyte ions. According to their results, mesoporous carbons had larger reversible capacity (384.4 mAh g$^{-1}$) and outstanding cycle performances as compared with polyacrylonitrile/lignin carbonized nanofibers (with reversible capacity of 304.4 mAh g$^{-1}$ at 20.0 mA g$^{-1}$). In another study, a novel nanocomposite based on zinc oxide/mesoporous carbon as the anode material of Li-ion batteries has been developed [88]. For this purpose, the researchers used polyvinyl alcohol as the carbon source and zinc nitrate hexahydrate to produce mesoporous structure through the precipitation method, followed by a calcination at 500 °C. They reported that zinc oxide nanoparticles with a diameter of 6.3 nm are uniformly anchored by the hydroxyl groups of polyvinyl alcohol. Zinc oxide/mesoporous carbon nanocomposites as the anode of a lithium-ion battery exhibited high reversible capacity (610 mAh g$^{-1}$) at 100 mA g$^{-1}$, which could be attributed to the synergistic effect of zinc oxide nanoparticles and the mesoporous carbon matrix. Raza et al. fabricated $SnO_2$ nanoparticles covered by mesoporous carbon structure derived from green microalgae as a template and carbon source through a hydrothermal process, followed by iron oxide etching [89]. They used

SnO$_2$ nanoparticles and prepared composite as an anode electrode material for lithium-ion batteries. According to their results, the SnO$_2$/mesoporous carbon composite showed a highly porous structure with a high surface area and a large pore volume compared with SnO$_2$ nanoparticles. This strategy improved the contact between the redox-active materials and the electrolyte in batteries (Table 4).

**Table 4.** Characteristics and electrochemical properties of SnO$_2$ nanoparticles and SnO$_2$/mesoporous carbon composite.

| Sample | Surface Area (m$^2$ g$^{-1}$) | Pore Volume (cm$^3$ g$^{-1}$) | Discharge Capacity (mAh g$^{-1}$) | Charge Capacity (mAh g$^{-1}$) |
|---|---|---|---|---|
| SnO$_2$ nanoparticles | 83.51 | 0.079 | 616.04 | 608.59 |
| SnO$_2$/mesoporous carbon | 798.9 | 1.17 | 1129.54 | 1107.09 |

Furthermore, SnO$_2$/mesoporous carbon was revealed to have a higher charge/discharge capacity than SnO$_2$ nanoparticles have. This can be attributed to the amorphous nature of mesoporous carbon and the high specific surface area of highly porous SnO$_2$/mesoporous carbon composite. In another study, mesoporous carbon with a graphitized carbon framework has been produced based on the template approach as anode materials for lithium-ion batteries [90]. The researchers used nano-SiO$_2$ powder as the hard template and mesophase pitch as the carbon precursor for the preparation of mesoporous carbon. They found that the mesoporous carbon materials with 10 wt% of SiO$_2$ as anode materials for lithium-ion batteries had a reversible capacity of 248.3 mAh g$^{-1}$ at 1 °C after 100 cycles (retention of 99.7%) and revealed an improved rate performance and good cycling stability. A new porous composite nanofiber has been prepared by Wang and et al. through electrospinning and foaming processes [91]. For this purpose, they used aluminum acetylacetonate as the foaming agent in nanofibers made of polyacrylonitrile/silicon composite. Mesopores were generated by foaming nanofibers through aluminum acetylacetonate sublimation. They used the produced mesoporous structure as anode in lithium-ion batteries after carbonization at 700 °C. Their results showed that the obtained mesoporous carbon materials had higher reversible capacity and better cycling retention than nonporous composites nanofibers (nanoporous polyacrylonitrile/silicon composite nanofibers) and carbon nanofibers (polyacrylonitrile naonfibers) alone (Table 5).

**Table 5.** Discharge capacity of various porous materials.

| Sample | Discharge Capacity in the 10th Cycle (mAh g$^{-1}$) | Discharge Capacity in the 20th Cycle (mAh g$^{-1}$) |
|---|---|---|
| Mesoporous polyacrylonitrile/silicon/aluminum acetylacetonate composite nanofibers | 1199 | 1045 |
| Nanoporous polyacrylonitrile/silicon composite nanofibers | 715 | 569 |
| Polyacrylonitrile nanofibers | 526 | 438 |

Although the mesoporous carbons show good performance as electrodes in Li-ion batteries, it is still required to find a simple, effective, and scaled-up synthetic strategy for the production of cheap and highly stable electrode materials. Thus, the future challenge is to synthesize highly pure and cheap mesoporous carbons or functionalized mesoporous carbons without any contamination, for practical applications.

*4.4. Fuel Cells*

A fuel cell, also known as a galvanic or voltaic cell, is an electrochemical device that can continuously convert the chemical energy of hydrogen and an oxidizing agent (often oxygen) into electricity through a pair of redox reactions by the production of water as the byproduct [92]. These energy-conversion devices can be used in the fields of portable, mobile, and stationary power. A key process in fuel cells is the oxygen-reduction reaction; its slow kinetics can limit the efficiency of energy conversion in a fuel cell. The platinum (Pt)-based electrodes have been regarded as the best electrocatalysts for oxygen-reduction reactions. However, a platinum-group metal-based electrode suffers from multiple disadvantages, such as high cost, scarcity, and poisoning by the oxidation of intermediates such as CO [93]. Numerous studies have been carried out to reduce or replace the Pt-based electrode with nonplatinum electrocatalysts in fuel cells. Mesoporous carbon materials with uniquely combined electrochemical and mass transport characteristics have garnered much attention as a new class of electrocatalysts for oxygen-reduction reactions with a low cost and a high efficiency for fuel cells. In this regard, several research efforts have been devoted to the development of fuel cells [94–103]. Viva et al. prepared mesoporous carbon with a pore distribution of about 20 nm and a pore volume of 0.99 cm$^3$ g$^{-1}$ and used it as support for platinum nanoparticles in fuel cells [99]. They found Pt catalysts supported on mesoporous carbon had a lower overpotential than Pt supported on Vulcan carbon. Their results showed that the use of mesoporous carbon-supported Pt nanocatalyst as a cathode catalyst for $O_2$ reduction had similar performance when compared with commercial catalyst, while there was an 8% improvement when used as anode catalyst for $H_2$ oxidation. In another study, platinum-supported ordered mesoporous carbon catalysts were synthesized by Kuppan et al. as an anodic catalyst for fuel-cell applications [100]. For this purpose, they used colloidal platinum reduced and deposited over ordered mesoporous carbon that was synthesized by silica hard template. They reported that the performance of prepared catalysts showed better activity and higher stability than commercial platinum-supported carbon catalysts. This could be attributed to the better dispersion of the platinum that could be obtained from a higher surface area and the large pore volume of the mesoporous carbon structures. Singh et al. fabricated mesoporous carbon nitride using carbon nitride and mesoporous silica material as a hard template [101]. Then, the obtained mesoporous structures were modified by immobilizing metal phthalocyanine, where metals included manganese, iron, cobalt, nickel, copper, and zinc. The resulting composites were evaluated for their electrocatalytic activity toward the oxygen-reduction reaction. They reported that iron-immobilized mesoporous carbon material exhibited a four-electron oxygen-reduction mechanism, higher oxygen-reduction reaction stability, and excellent methanol tolerance in a basic medium when compared with other composites and platinum materials. Thus, the iron-immobilized mesoporous carbons might be considered as a promising cathode material for methanol fuel cells. In another work, Chen et al. utilized a microporous mesoporous carbon derived from chestnut shell as the anode material in microbial fuel cells [102]. They reported that mesoporous carbon structures were synthesized via a carbonization procedure, followed by a chemical activation process. They found that a chemical activation process could reduce the O-content and N-content groups on the chestnut shell anode. According to Table 6, a fuel cell with activated carbon showed higher maximum power density when compared with a carbon cloth anode, which might be attributed to the higher specific area after activation and less oxygen-containing surface functional groups that are associated with produced microporous and mesoporous structures.

**Table 6.** Power density of microbial fuel cells with different anodes.

| Sample | Power Density (Wm$^{-3}$) |
|---|---|
| Mesoporous carbon | 23.6 |
| Carbon cloth | 10.4 |

Xie et al. investigated the effect of introducing mesoporous carbon into an oxygen-reduction reaction as a cathode catalyst support in fuel cells [103]. Their results showed that platinum nanoparticles with an average diameter of 2.8 nm were distributed on the surface of carbon and could be deposited in the mesoporous inside the support. They found that fuel cells based on mesoporous carbon structure showed excellent performance and improved oxygen-reduction reaction activity for the catalyst. Many efforts have been made in the field of using mesoporous carbons in fuel cells, but the catalyst synthesis methods using mesoporous carbon support should be developed to support platinum-based nanocatalysts with a loading level above 90 wt% while maintaining a small particle size.

*4.5. Biosensors*

Biosensors are devices that contain a biological component for analyte detection and a physicochemical component for signal generation. The detection of large molecules such as nucleic acids and proteins by biosensors faces significant problems, such as electrode fouling, the nonspecific adsorption of biological components at the electrode surface, and a lack of sensitivity in the appropriate concentration range. Recently, nanostructure electrode materials have attracted great interest because of the better electrochemical performance as compared with traditional materials. The unique characteristics of mesoporous carbon materials, including their biocompatibility, the large conducting surface area, and their regular and widely open nanostructure, make them suitable materials for the preparation of electrodes for bioelectrochemical sensor applications [104]. In this regard, a lot of work has been conducted [105–112]. Zhong et al. fabricated the enzyme-modified electrode of electrochemical biosensor based on a complex enzyme that consisted of a mesoporous carbon/zirconium-based metal organic framework composite with embedded laccase for tetracycline detection [108]. They indicated that the presence of a mesoporous structure with a high specific surface area in an electrode was beneficial to enzyme immobilization and the protection of the laccase against inactivation and denaturation. The superior conductivity of the final composite improved the electron transfer in the modified electrode. They found that the biosensor based on this complex enzyme showed a relatively low detection limit ($8.94 \times 10^{-7}$ mol $L^{-1}$) and good stability and reproducibility compared with that made using pure laccase for tetracycline detection.

The detection of the level of a reactive oxygen species such as hydrogen peroxide ($H_2O_2$) is very important because of the far-ranging impacts of $H_2O_2$ homeostasis on human health. On this subject, Wang et al. prepared CdZnSeS quantum dots (QDs) condensed with ordered mesoporous carbon (OMC) and then immobilized them onto a glassy carbon electrode surface to detect hydrogen peroxide in biosensors [109]. They reported that a combination of QDs with the highly conductive mesoporous carbons structures enhanced the electrochemical performance and electrochemical luminescence signal. They found that the produced biosensor had a faster electron transfer rate, a lower detection limit, desirable selectivity, and a long-time stability as compared with a glass carbon electrode. In another study, Jia et al. synthesized mesoporous carbon and used it for the selective determination of dopamine [110]. For this purpose, mesoporous carbon was produced by using mesoporous silica materials SBA-15 as the template and sucrose as the carbon source to modify the surface of a glass carbon electrode for applications in biosensors. The presence of a mesoporous structure improved the electron transfer rate as compared with a glass carbon electrode. This could be attributed to a higher electrochemical surface and larger number of edge plane defect sites on the mesoporous carbon materials. According to their results, the mesoporous carbon-modified electrode revealed high electrocatalytic activities toward the oxidation of dopamine and ascorbic acid and displayed a desirable level of the selective electrochemical determination of dopamine in the presence of ascorbic acid. Song et al. modified a glass carbon electrode by mesoporous carbon and functionalized mesoporous carbon with carboxyl and amino groups for the selective determination of dopamine in the presence of ascorbic acid [111].

The researchers [111] synthesized mesoporous carbon by replacing mesoporous silica SBA-15 using sucrose as the carbon source and then by reacting it with concentrated nitric acid and ethylenediamine to produce carboxyl-modified and amine-modified mesoporous carbon, respectively (Figure 6).

**Figure 6.** Surface functionalization of mesoporous carbons with concentrated nitric acid and ethylene-diamine.

They reported that high surface areas and the presence of mesopores structure were beneficial to enhancing the selectivity and sensitivity of the determination of dopamine in biosensors (Table 7).

**Table 7.** The characteristic of modified electrodes.

| Sample | Detection Limit (µM) |
|---|---|
| Mesoporous carbon-modified glass carbon electrode | 0.0045 |
| Mesoporous carbon (with carboxyl groups)-modified glass carbon electrode | 0.044 |
| Mesoporous carbon (with amino groups)-modified glass carbon electrode | 0.33 |

In another work, Balasubramanian et al. prepared boron- (B) and nitrogen (N)-codoped mesoporous carbon and utilized them for modification of the carbon electrodes in sensors to detect isoniazid [112]. They reported that B/N-doped mesoporous carbon-modified electrode had an excellent electrochemical performance toward isoniazid detection with a broad dynamic range (0.02–1783 µM) and a low detection limit (1.5 nM). This could be attributed to the mesoporous structural architecture, large surface area, low pore size, and high electrochemical conductivity. Although many studies have been conducted in the field of using mesoporous carbons in sensors, there are still challenges. One of the challenges is that we do not exactly know the electrochemical behavior of carbons. There are many factors affecting the efficiency of a mesoporous carbon-based electrode that must be individually understood. For example, the recognition of the observed electrocatalytic effects, which is different from one material to another material, helps in choosing the suitable mesoporous carbon for the intended applications. Another problem is that because of the powdery nature of the mesoporous carbons, an additional binder is needed to immobilize mesoporous carbons on the surface of an electrode. To solve this problem, it is necessary to conduct studies on the direct production of thin films of ordered mesoporous carbon materials onto the electrode surface.

### 4.6. Catalysis

Catalysis is the process of increasing the rate of a chemical reaction by adding catalyst. Catalysts are necessary for various processes that can shorten chemical process routes and improve sustainability. In various industries, the practical application of liquid catalysts is limited because of corrosion and environmental problems. Thus, many attempts have been made to replace the liquid catalysts with solid catalyst because of the easy product separation and for recycling. Many types of solid catalysts have been developed, the most important of which are catalysts based on metals and their alloys, organic nitrogen–containing compounds, alkali metal hydroxides metallic oxide, zeolites, active carbon, and ion-exchangeable resins. Two of the efficient metal catalysts are platinum (Pt) and a platinum-based alloy. However, the commercialization of Pt-based catalysts is hindered by the limited reserves on the earth, their high cost, and the instability in the presence of carbon monoxide and methanol, especially in fuel cells [92]. Furthermore, organic nitrogen–containing compounds such as pyridine, piperidine, alkali metal hydroxides, metallic oxide (such as sodium hydroxide and potassium hydroxide) have major disadvantages. The major drawback of them is the difficulty in catalyst/product separation and catalyst recycling owing to their homogeneous nature. On the other hand, heterogeneous metal oxides catalysts, including magnesium aluminum oxide and magnesium oxide, show good catalyst recyclability but suffer from metal contamination in the reactions [50,113]. In general, many of the commercial catalysts used have a series of disadvantages, such as high reaction temperature, low efficiency, low region-selectivity, and nonreutilization. In recent decades, mesoporous carbons have attracted much attention for catalyst applications thanks to their promising structural features, including unique optoelectronic properties, high porosity, excellent electron conductivity, high surface area, high chemical and physical stability, large pore volume, and biocompatibility [114]. Extensive studies have been done in the field of using mesoporous carbon as catalysts or as catalyst supports [113–122]. On this subject, Morales-Acosta et al. prepared platinum nanoparticles deposited on ordered mesoporous carbon to replace Vulcan XC-72 in fuel0-cell applications [118]. For comparison, they also deposited platinum nanoparticles on the multiwalled carbon nanotubes for methanol, ethanol, and ethylene glycol electro-oxidation. Their results demonstrated that platinum/mesoporous carbon improved the catalytic activity toward the three reactions. This could be attributed to the mesoporous structure of carbon, which improved support–catalyst interactions. In another work, Hwang et al. fabricated mesoporous carbon by using the hard-template nanocasting method and applied it as a support material of Fe catalysts for the production of liquid fuels from the direct hydrogenation of carbon dioxide [119]. According to their results, mesoporous structures revealed excellent catalytic activity and selectivity toward long-chain hydrocarbons in direct carbon dioxide hydrogenation. They found that the mesoporous structure could improve the mass transfer of hydrocarbon molecules, which resulted in the enhancement of carbon dioxide conversion and the selectivity of hydrocarbons with more than five carbon atoms. Xu et al. prepared magnetic mesoporous carbon and applied it as an efficient heterogeneous Fenton catalyst [120]. For this purpose, the researchers [120] used waste-lignin and ferric chloride as precursors and nitrobenzene as a model pollutant to evaluate the catalytic ability of the mesoporous structure as a heterogeneous Fenton catalyst. They reported that a mesoporous structure with an average pore size of 3.86 nm and a specific surface area of 138 $m^2$/g had higher catalytic efficiency than iron oxide. Nitrogen and sulfur dual-doped, metal-free mesoporous carbon was prepared as a cathodic catalyst in a direct biorenewable alcohol fuel cell by Qiu et al. [121]. For this purpose, they used glucose as the carbon source and ammonia and thiophene as nitrogen and sulfur precursors, respectively, to produce a mesoporous structure with a high surface area (1023 $m^2$ $g^{-1}$). They found that the prepared mesoporous at 800 °C showed the highest oxygen-reduction reaction activity with the onset potential of 0.92 V vs. RHE, Tafel slope of 68 mV $dec^{-1}$. Furthermore, the mesoporous structure showed 88.2 mW $cm^{-2}$ peak power density, which represented 84% of the initial performance of the one with a platinum/Vulcan-carbon cathode. They indicated that a fuel cell with a

mesoporous structure as a cathode could maintain a peak power density at 90.6 mW cm$^{-2}$ after two hours of operation, while the platinum/Vulcan-carbon cathode-based fuel cell lost 36.9% of its peak power density. The improvement of the activity of the mesoporous structure as a catalyst could be attributed to the synergistic effect of graphitic-N and S atoms. In another report, Parthiban et al. synthesized mesoporous carbon through a soft-template method using a triblock copolymer (Pluronic F-127) [122]. They incorporated fluorine atoms into porous carbon by using ammonium fluoride. They reported that the incorporation of fluorine created defects in the matrix structure, which was beneficial for oxygen-reduction reactions. They claimed that the fluorine-doped mesoporous carbons revealed good catalytic activity and long-term electrochemical stability. However, one of the most important challenges of using of mesoporous carbon materials as catalytic supports is uncontrollable doping and irreversible agglomeration at the molecular level during high-temperature pyrolysis in their synthetic technique. Furthermore, it is still necessary to invent a cost-effective approach for the commercialization and large-scale synthesis of mesoporous carbons.

### 4.7. Enzyme Immobilization

Enzyme immobilization is defined as the imprisonment of enzyme molecules onto a matrix, conducted physically or chemically, so that it can maintain its full activity or most of its activity. In this regard, many studies have been conducted on the use of mesoporous carbons as a matrix for enzyme immobilization [123–131]. Zhu et al. prepared mesoporous carbons and then glucose oxidase incorporated in mesoporous structure by mechanical mixing to prepare the electrode of sensitive and selective glucose biosensors [127]. They compared mesoporous-based electrodes with that constructed of carbon nanotubes and glucose oxidases. They reported that the mesoporous carbon/glucose oxidase electrode had a larger current signal with a sensitivity of 6.3-fold of that of a carbon nanotube/glucose oxidase electrode. Furthermore, they indicated that the use of a mesoporous-structure-based electrode could decrease the detection limit over than of a carbon nanotube/glucose oxidase electrode (Table 8).

**Table 8.** Characteristic of the prepared electrodes for a glucose biosensor.

| Sample | Current Signals ($\mu$A M$^{-1}$) | Detection Limit (mM) |
| --- | --- | --- |
| Mesoporous carbon/glucose oxidase | 18.42 | 0.13 |
| Carbon nanotube/glucose oxidase | 2.92 | 0.072 |

In another study, Dascalescu et al. prepared a laccase-based biosensor for the detection of serotonin in food supplements [128]. They immobilized an enzyme on a mesoporous carbon-modified carbon screen-printed electrode. They reported that a modified electrode showed a detection limit value of 316 nM. The mesoporous carbon/laccase-based sensor showed higher efficiency than an unmodified mesoporous carbon-based sensor. Yu et al. deposited platinum nanoparticles on mesoporous carbons and then glucose oxidase immobilized in the platinum/mesoporous carbon matrix [129]. Mesoporous carbons were synthesized by mesoporous silica material SBA-15 as a template and sucrose as a carbon source. They indicated that the biosensor revealed excellent current response to glucose and good catalytic activity. Furthermore, the biosensor showed high thermal stability and long-time stability, which could be related to the protection effect of the mesoporous carbon matrix. In another work, the researchers immobilized laccase on to carbon-based mesoporous magnetic composites and then applied it for the removal of phenolic compounds [130]. Their results showed a high adsorption capacity (491.7 mg g$^{-1}$) with good activity recovery (91.0%) and broader pH and temperature profiles than free laccase. Furthermore, the thermal and operational stability was improved. Kim et al. fabricated glucose-sensing electrodes from mesoporous electro-spun carbon fibers acti-

vated by potassium carbonate [131]. The presence of the porosity and oxygen functional groups on the carbon fibers enhanced the immobilization efficiency of the glucose oxidase enzyme. The sensitivity of the electrodes prepared from an activated mesoporous carbon was 3.4 μA mM$^{-1}$ cm$^{-2}$, which was higher than a nonactivated carbon fibers electrode. Although many efforts have been made to immobilize the enzymes on mesoporous carbon structures, designing a good structure and finding a suitable immobilization method in order to increase the stability and activity of enzymes remains an important challenge. Thus, more studies should be conducted in this field.

*4.8. Drug Delivery*

Drug delivery is a process in which carrier systems are used for effective therapeutic pharmaceuticals delivery [132]. The design of carbon-based formulations for the encapsulation of drugs suggests alternative pharmacotherapy options with improved safety profiles for current drugs [133]. Recently, mesoporous carbons have attracted much attention as therapeutic agent carriers thanks to their large surface area, adjustable and narrow mesopore distribution, high pore volume, and good biocompatibility. In this regard, many studies have used mesoporous carbons as drug carriers [21,134–141]. Wang et al. prepared a drug carrier based on nitrogen-doped mesoporous carbon nanoparticles using chitosan as a carbon and nitrogen source and a triblock copolymer (F127) as a soft template with the coupling of spray drying and pyrolysis methods [138]. They modified mesoporous carbons through oxidation by ammonium persulfate to improve the hydrophilicity of the carbon materials. They reported that the specific surface area, pore volume, and contact angle of oxidized mesoporous carbons reduce to a certain extent as compared with unmodified mesoporous structure thanks to the introduction of a large number of oxygen-containing functional groups (Table 9).

**Table 9.** The main characteristics of different carriers for hydroxycamptothecin.

| Sample | Surface Area (m$^2$/g) | Pore Volume (cm$^3$/g) | Contact Angle | Drug Loading (mg/g) | Release after 12 h (%) |
|---|---|---|---|---|---|
| Mesoporous carbon | 804 | 0.87 | 133.4° | 676.97 | 83.40 |
| Oxidized mesoporous carbon | 322 | 0.64 | 58.2° | 647.20 | 81.11 |

Their results showed that oxidized mesoporous carbon materials had a lower adsorption capacity for hydrophobic hydroxycamptothecin than that of the mesoporous carbons and attributed it to the smaller surface area. Furthermore, the release rate of row drug was improved by the introduction of mesoporous carbon materials. In another study, a drug delivery system based on polyethylene glycol-modified oxidized mesoporous carbon with a narrow size distribution of 90 nm was fabricated for the delivery of doxorubicin [139]. For this purpose, mesoporous carbon was synthesized through the hydrothermal approach and then modified by hydrogen peroxide and polyethylene glycol-modified phospholipid (DSPE-mPEG) (Figure 7). They indicated that prepared nanocarriers showed good stability under neutral pH conditions while releasing the drug in an acidic tumor environment. Furthermore, they showed that mesoporous nanostructures efficiently penetrated the membrane of tumor cells and inhibited the growth of cancer cells both in vitro and in vivo.

**Figure 7.** Synthesis of drug-loaded mesoporous carbons.

Wan et al. functionalized mesoporous carbon nanoparticles with hyaluronic acid for enzyme-responsive drug delivery [140]. They reported that grafting of hyaluronic acid via an electrostatic interaction resulted in stable drug encapsulation in mesoporous carbon and in an improvement in cellular uptake in cancer cells overexpressing CD44 receptors. Furthermore, nanocarriers loaded with doxorubicin and verapamil showed a dual pH and hyaluronidase-1 responsive release in the tumor. The IC50 of mesoporous loaded drugs (0.323 $\mu$g mL$^{-1}$) was lower than DOX/VER (0.978 $\mu$g mL$^{-1}$) on HCT-116 cancer cells, which could be attributed to the presence of the mesoporous structure. They found that a prepared mesoporous structure exhibited a superior therapeutic effect on colorectal tumors in BALB/c nude mice. In another work, Tian et al. made a drug delivery system based on mesoporous carbons functionalized with arginine-glycine-aspartic acid (RGD) for doxorubicin delivery and chemo-photothermal therapy for prostate cancer [141]. The as-prepared nanocarriers showed a high drug-loading efficiency of 246 $\mu$g/mg, good heat transformation ability, and an excellent pH/near-infrared (NIR) dual stimuli-responsive release. They indicated that the therapeutic efficacy of nanocarriers toward PC3 cells upon NIR laser irradiation had been improved. Zhang et al. synthesized a pH and redox responsive drug delivery system based on polyacrylic acid–capped mesoporous carbon nanoparticles for the delivery of doxorubicin in cervical cancer treatments [21]. For this purpose, polyacrylic acid was conjugated to the surface of mesoporous carbons through disulfide bonds (Figure 8). They reported that acidic environment of tumor tissues induced the partial release of the drug from the surface of nanocarriers. Furthermore, the presence of higher concentrations of glutathione in tumor sites led to dissociation of disulfide bonds and resulted in the greater release of doxorubicin. Their cellular cytotoxicity test showed good biocompatibility for mesoporous nanocarriers and a high therapeutic effect for HeLa cells.

The biomedical applications of mesoporous carbons are still in the early stage. There still remain several significant challenges facing the use of mesoporous carbon materials for disease treatments. For example, the in vivo real-time monitoring of mesoporous carbons vehicles via a bioimaging method needs to be developed. More detailed knowledge about drug delivery systems based on mesoporous carbon materials is needed to develop more-comprehensive therapeutic techniques. In addition, most of the efforts in the toxicological data are based on in vitro tests and our knowledge about the in vivo fates is limited because of the difference between the in vitro and in vivo results.

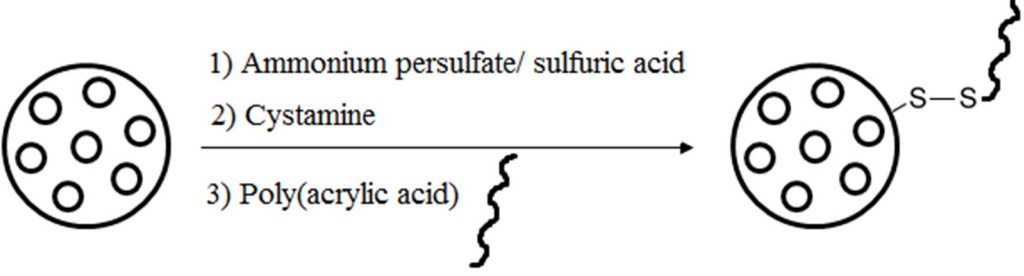

**Figure 8.** Surface modification of mesoporous carbons with poly(acrylic acid).

*4.9. Other Applications*

Water is essential for the survival of life in all forms. Unfortunately, a variety of released pollutants from industrial activities has resulted in poor water quality. In general, water pollutant materials include inorganic pollutants such as heavy metals, anions, and organic pollutants such as dyes, phenolic compounds, proteins, and vitamins [142]. The presence of these pollutants, especially organic dyes, reduces the penetration of sunlight and limits the process of photosynthesis. Several methods have improved the removal of aquatic pollution, such as biological treatment, membrane technology, ion exchange, ozonation, and adsorption [143]. Adsorption with mesoporous carbon materials has attracted much attention in recent years. On this subject, Ezzeddine et al. indicated that mesoporous carbons showed excellent adsorption (1250 mg g$^{-1}$) of methyl blue [144]. Studies showed that surface modification can improve mesoporous carbons absorption capacity. Guo et al. claimed that the oxidization of mesoporous carbons with nitric acid increased the adsorption capacity of resorcinol from 36.3 to 39.2 mg g$^{-1}$ thanks to the incorporation of oxygen-containing groups [145]. Although many studies have shown the suitability of mesoporous carbons for the removal of organic and inorganic pollutants from the aqueous phase, there is not enough information about the exact mechanism of the absorption of different pollutants by mesoporous carbon materials. In general, the removal of pollutants by mesoporous carbons is significantly influenced by the physicochemical properties of ordered mesoporous carbons, such as specific surface area, pore size, pore volume, and functional groups, as well as the physicochemical properties of pollutants, including molecular size, surface charge, functional groups, and hydrophobicity. Furthermore, adsorption conditions such as solution pH, initial solution concentration, temperature, and contact time are important concepts for better understanding of the adsorption mechanism. Therefore, many studies must be conducted to better understand the absorption mechanism and consequently for the selection of suitable mesoporous carbon structures to remove any type of pollution. The limited resources of fossil fuels dictate developing the use of environmentally friendly energy carriers such as hydrogen. Hydrogen (H$_2$) is an ideal and pollution-free energy source. However, hydrogen storage is a major problem in the development of hydrogen-based technologies such as fuel cells [146]. Recently, mesoporous carbons have been shown as suitable candidates for H$_2$ storage owing to their large specific surface area, high porosity, and tunable pore structures. Hydrogen storage can be carried out in two ways: (a) as chemical hydrides or complex hydrides, such as NaAlH$_4$, NH$_3$, BH$_3$, and MgH$_2$, or (b) physically, in which, high pressure encapsulates H$_2$ in mesoporous carbons with a large adsorption surface [11,146]. For example, Wang et al. prepared nitrogen-doped mesoporous carbons for hydrogen storage. For this purpose, they synthesized mesoporous carbon with a high surface area (1362–3009 m$^2$·g$^{-1}$) through the hydrothermal carbonization and chemical activation of chitosan as a carbon source. Their results showed that porous carbon offered excellent performance for hydrogen storage of 6.77 wt% at 20 bar, 77 K. [147]. The most important problem in this field is that the hydrogen adsorption was limited by the adsorbent density, the pore structure of the adsorbent, and the pore volume of the narrowest pores. Materials

with high porosity do not necessarily adsorb much hydrogen, which may be related to the lower interaction energy of hydrogen in wide pores compared with smaller micropores. Pores with a diameter of less than 1 nm are most efficient for hydrogen storage rather than for mesopores [148]. On the other hand, mesoporous carbons can also be used for the detection of various materials in different industries thanks to their high surface area, large pore volume, and high electrical conductivity. Yu et al. [149] showed that mesoporous carbons can detect two dihydroxybenzene isomers (catechol and hydroquinone) better than multiwalled carbon nanotubes (MWCNTs) and Vulcan XC-72 carbon. The sensitivity for catechol and hydroquinone was 41 $\mu A/cm^2$ $\mu M$ and 52 $\mu A/cm^2$ $\mu M$, respectively. The detection of Cd(II), Cu(II), Hg(II), and Pb(II) metal ions by square-wave anodic stripping voltammetry has been studied in the presence of oxidized ordered mesoporous carbon decorated with magnetite [detection-1]. The results showed high selectivity toward Pb(II) with a favorable sensitivity 145.75$\mu A$ $mg^{-1}$ L (30.19 $\mu A/\mu M$) and a detection limit of 1.57 $\mu g$ $L^{-1}$ (7.57 nM), while the detection limits for Cd(II), Cu(II), and Hg(II) ions were 0.059, 0.0024, and 0.167 mg $L^{-1}$, respectively. The presence of magnetic properties increased the sensitivity for Pb(II) detection and maintained the capacity of the simultaneous detection for all the metal ions. Surface modification of mesoporous carbon materials can improve their detection efficiency [150]. One of the important problems in this field is the production of functionalized mesoporous carbons with a high surface area, which is critical for very sensitive detections. Ultrafiltration membranes are a kind of membrane with average pore diameters of 1–100 nm, and they are used for industrial separation processes. Generally, membranes used for ultrafiltration are made from polymer. Poor chemical stability and thermal stability limit the application of these membranes [151,152]. Recently, the use of mesoporous carbons in ultrafiltration membranes have attracted significant attention in a variety of industrial process applications thanks to their advantages, such as high thermal and chemical stability. For example, Wei al. prepared mesoporous carbon membranes with an average pore size of 7.1 nm through sol-gel synthesis, followed by supercritical drying and carbonization, and used formaldehyde and resorcinol as an original precursor [151]. The pure water flux was 13.4 L·$m^{-2}$·$h^{-1}$·$bar^{-1}$, and the molecular weight cutoff was about 2000. Although mesoporous carbon membranes attracted much attention, the complex preparing process and low pure water flux still need to be improved. Thus, further study must be conducted in this field so that their use in purification and separation processes can expand. It is known that the direct thermal ablation of cancer cells needs a high temperature, which can damage the normal cells. Therefore, photothermal therapy is widely applied to treat cancer by using near-infrared resonant nanoagents, which can convert the infrared light into heat and induce the ablation of cancer cells. Mesoporous carbon materials show strong optical absorption in the near-infrared region (808 nm), indicating their potential utility as a photothermal agent. Xu et al. recently revealed that folic acid–targeted mesoporous carbon materials had superior photothermal conversion efficiency than graphene oxide [135]. Furthermore, they indicated that infrared irradiation facilitated the release of loaded anticancer drugs from mesoporous carbons. In addition, the use of mesoporous carbon materials in bioimaging have been studied in recent years. For instance, the MRI imaging of mesoporous carbon can be performed by embedding certain inorganic nanoparticles, such as $Fe_3O_4$ and manganese oxide, into the carbonaceous structure. Zhang et al. developed an intelligent nanosystem based on mesoporous carbon structures to enhance the resolution and specificity of diagnostic imaging and improve therapeutic efficiency for cancer treatment [153]. However, scalable synthetic methods for mesoporous carbons with optimized structural and compositional parameters for biological applications are necessary. Thus, one of the challenging problems is that general, controllable, and standard procedures for obtaining mesoporous carbon materials especially for size-tunable of them have not yet been developed. Furthermore, systematic biosafety evaluations should be conducted to guarantee their clinical translation in the future, and these evaluations depend on the development of procedures to achieve desirable mesoporous carbon materials.

## 5. Conclusions and Future Outlook

The presented review attempted to provide comprehensive information about synthetic strategies for mesoporous carbon materials, along with their various applications. Synthetic methods for mesoporous carbons, such as physical and chemical activation, hard template, soft template, the carbonization of polymer/polymer blends, organic aerogels, and catalytic activation using metal ions, were discussed in detail. In sum, mesoporous carbon materials were considered as the next generation of inorganic materials for various applications thanks to their unique mesoporous structure, carbonaceous composition, and high biocompatibility. The recent studies on the various applications of mesoporous carbon materials, including as adsorbents; as electrode materials in supercapacitors, lithium-ion batteries, proton exchange membrane fuel cells, and biosensors; as catalyst supports in different reactions; as hosts for enzyme immobilization; and as nanocarriers in drug delivery, were also discussed. Although various studies revealed that mesoporous structures were suitable materials for the mentioned applications, they are still at the very early stage of research. More studies are required to find efficient and inexpensive preparation methods for mesoporous carbons for industrial and social applications on a large scale. Furthermore, the use of mesoporous carbon materials in biomedical applications such as diagnostic imaging is rare compared with mesoporous silica nanoparticles and other carbon nanomaterials. In addition, biosafety evaluations of mesoporous carbon structures are important in the field of biological applications. According to the mentioned challenges, combining existing synthetic methods may be a subtle approach for the practical production of mesoporous carbon materials to lower the consumption of consumables and energy and to decrease waste. On the other hand, it has been shown in some studies that increasing the pore volume can facilitate the immobilization of molecules within the pores. So it is important to explore the possibilities of the synthesis of mesoporous carbons with large pore sizes in order to incorporate large molecules within the pores. In addition, it is interesting to investigate the factors that affect the combination of the properties of the incorporated materials and those of mesoporous carbons. Because more than 95% of the studies have focused on a single mesoporous carbon type, it can be interesting to investigate the use of several materials at the same time to show distinct structural and porosity characteristics. Therefore, in the near future, given the outstanding properties of mesoporous carbons, it will be expected that with development of material, chemistry, and nanoengineering science, researchers will be able to design suitable mesoporous carbon materials with excellent abilities for various applications.

**Author Contributions:** Conceptualization, S.M.-A. and E.A.; methodology, S.M.-A.; software, E.A.; validation, E.A.; investigation, E.A.; resources, E.A.; data curation, E.A.; writing—original draft preparation, E.A.; writing—review and editing, S.M.-A. and E.A.; visualization, S.M.-A.; supervision, S.M.-A.; project administration, S.M.-A. All authors have read and agreed to the published version of the manuscript.

**Funding:** This research received no external funding.

**Conflicts of Interest:** The authors declare no conflict of interest.

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
