# Peer review of "Mesoporous Carbon-Based Materials: A Review of Synthesis, Modification, and Applications"

_catalysts, doi:10.3390/catal13010002_

Round 1

Reviewer 1 Report

The manuscript Synthesis and applications of mesoporous carbon-based materials is an interesting article and deserve publication, however following comments should be addressed/incorporated before publication

Title should be modified, it should be more self-explanatory, Rewrite the introduction with emphasis on research background and objectives.    Add some information about the structural properties of mesoporous carbon materials. Add a section regarding modification of mesoporous carbons and how it can improve the properties and applications of mesoporous carbon.

Format the caption of Table 3

Correct the Format of Table 4

Format the caption of Table 5

There are many grammatical mistakes throughout the text. Please recheck the  English through out the manuscript.

Author Response

Reviewer 1:

The manuscript Synthesis and applications of mesoporous carbon-based materials is an interesting article and deserve publication, however following comments should be addressed/incorporated before publication

  1. Title should be modified, it should be more self-explanatory

Response: Based on the reviewer’s comment, the title was modified as follows: “Mesoporous carbon-based materials: a review of synthesis, modification, and applications“.

  1. Rewrite the introduction with emphasis on research background and objectives.  Add some information about the structural properties of mesoporous carbon materials.

Response: The authors have tried their best to express the research background, their objectives and the information about the structural properties of mesoporous carbon materials within the introduction (page 1,2,3,4) section. For more clarification the background and objectives of this work and the information about the structural properties of mesoporous carbon materials were re-written.

  1. Add a section regarding modification of mesoporous carbons and how it can improve the properties and applications of mesoporous carbon.

Response: Based on the reviewer’s comment, a section regarding modification of mesoporous carbons was added into manuscript on page 8 and 9.

  1. Format the caption of Table 3

Response: Based on the reviewer’s comment, the caption of Table 3 was corrected.

  1. Correct the Format of Table 4

Response: Based on the reviewer’s comment, the format of Table 4 was corrected.

  1. Format the caption of Table 5

Response: Based on the reviewer’s suggestion, the caption of Table 5 was corrected.

  1. There are many grammatical mistakes throughout the text. Please recheck the  English through out the manuscript.

Response: The manuscript has thoroughly double-checked in terms of English language structure and the sentences were polished. The corrections were all highlighted in red colour.

Reviewer 2 Report

Recommendation: Publish after minor revisions noted.

Comments:
In this review, the authors discussed the synthesis and application of mesoporous carbon-based materials. The manuscript is well written and generally clear in its presentation.  The comments and concerns that I have should be straightforward for the authors to address.  Specifically,

Detailed comments are listed below:

1)    The authors introduce the application of mesoporous carbon-based materials in many different fields. It is better if they can discuss the challenges in each field.

2)    The unique materials are always interesting to the field of catalysis, the authors may need to expand the discussion of this material in the section of catalyst.

3)    Line 189: Citations are needed.

4)    Line 512: Citations are missing.

5)    In the section of conclusion and future outlook, it would be better if the authors include more of their thoughts on this type of materials.

Author Response

Reviewer 2:

Recommendation: Publish after minor revisions noted.

Comments:
In this review, the authors discussed the synthesis and application of mesoporous carbon-based materials. The manuscript is well written and generally clear in its presentation.  The comments and concerns that I have should be straightforward for the authors to address.  Specifically,

Detailed comments are listed below:

  1. The authors introduce the application of mesoporous carbon-based materials in many different fields. It is better if they can discuss the challenges in each field.

Response: In respect of the reviewer’s comment, various applications of mesoporous carbon-based materials and their challenges were added to the manuscript on page 20,21,24,26,28, 31, 33, 34,37,38,39,40.

  1. The unique materials are always interesting to the field of catalysis, the authors may need to expand the discussion of this material in the section of catalyst.

Response: According to the reviewer’s comment, the unique materials for catalyst was discussed in the section of catalyst on page 31.

  1. Line 189: Citations are needed.

Response: According to the reviewer’s comment, citation was added on page 21.

  1. Line 512: Citations are missing.

Response: According to the reviewer’s comment, citation was added on page 32.

  1. In the section of conclusion and future outlook, it would be better if the authors include more of their thoughts on this type of materials.

Response: In respect of the reviewer’s suggestion, the authors thoughts were added in the section of conclusion and future outlook (page 41).

Reviewer 3 Report

- Table 1. : reorganise the applications of mesoporous carbons according to the synthetic methods.

- Check the font size of table's titles and lines. Ex: Lines 531-539, ....

- The reference numbers are duplicated.

- Improve the quality of figures, schemes and tables.

- Authors need to add figures to explain each method used to prepare mesoporous carbons.

- Other applications of mesoporous carbon-based materials should be mentioned in the review: aquatic pollution, hydrogen storage, detection, ultrafiltration membranes,......

- Authors need to update the references of this manuscript. Several papers about mesoporous carbon-based materials were published in 2022. Examples of references to be quoted:
       - https://doi.org/10.1002/admi.202101998
       -  https://doi.org/10.1002/smsc.202200045
       - https://doi.org/10.1002/admt.202200237
       - https://doi.org/10.1021/acs.chemmater.8b03345
      - http://dx.doi.org/10.24294/can.v5i2.1700
      - https://doi.org/10.1002/smll.202202238
      -  https://doi.org/10.1002/admi.202101964
      - https://doi.org/10.1002/adfm.202209201
      - https://doi.org/10.1002/adfm.202208349
      - https://doi.org/10.1002/advs.202105603
     - https://doi.org/10.1155/2022/4949916
     -  https://doi.org/10.1002/idm2.12008
     - https://doi.org/10.1002/smll.202200326
    - https://doi.org/10.1155/2022/7333005
    - https://doi.org/10.1002/adfm.202107166
   -  https://doi.org/10.1002/9783527828562.ch3
   - https://doi.org/10.1016/S1872-5805(22)60577-8
   - .......

Author Response

Reviewer 3:

  1. Table 1. : reorganise the applications of mesoporous carbons according to the synthetic methods.

Response: In respect of the reviewer’s suggestion, Table 1 was reorganised.

  1. Check the font size of table's titles and lines. Ex: Lines 531-539, ....

Response: In respect of the reviewer’s suggestion, font size of table's titles and lines were checked.

  1. The reference numbers are duplicated.

Response: In respect of the reviewer’s comment, the reference numbers were checked and corrected.

  1. Improve the quality of figures, schemes and tables.

Response: figures and tables were replaced with high quality ones.

  1. Authors need to add figures to explain each method used to prepare mesoporous carbons.

Response: In respect of the reviewer’s comment, Figure 1 and 2 was added to the manuscript on page 5 and 7.

  1. Other applications of mesoporous carbon-based materials should be mentioned in the review: aquatic pollution, hydrogen storage, detection, ultrafiltration membranes, …

Response: In respect of the reviewer’s suggestion, other applications of mesoporous carbon-based materials was added to the manuscript. Other applications section was added to the manuscript on page 37.

  1. Authors need to update the references of this manuscript. Several papers about mesoporous carbon-based materials were published in 2022. Examples of references to be quoted:
    - https://doi.org/10.1002/admi.202101998
    -  https://doi.org/10.1002/smsc.202200045
           - https://doi.org/10.1002/admt.202200237
           - https://doi.org/10.1021/acs.chemmater.8b03345
          - http://dx.doi.org/10.24294/can.v5i2.1700
          - https://doi.org/10.1002/smll.202202238
          -  https://doi.org/10.1002/admi.202101964
          - https://doi.org/10.1002/adfm.202209201
          - https://doi.org/10.1002/adfm.202208349
          - https://doi.org/10.1002/advs.202105603
         - https://doi.org/10.1155/2022/4949916
         -  https://doi.org/10.1002/idm2.12008
         - https://doi.org/10.1002/smll.202200326
        - https://doi.org/10.1155/2022/7333005
        - https://doi.org/10.1002/adfm.202107166
       -  https://doi.org/10.1002/9783527828562.ch3
       - https://doi.org/10.1016/S1872-5805(22)60577-8

Response: The mentioned references were added to the text.(page 1,2,3,9)

Round 2

Reviewer 1 Report

The author has incorporated the suggested changes, however grammar and some minor typo mistakes needs to recheck.

I feel pleasure to accept the manuscript

Author Response

Q: The author has incorporated the suggested changes, however grammar and some minor typo mistakes needs to recheck. I feel pleasure to accept the manuscript.

Answer: The grammar and some minor typo mistakes were rechecked and corrected.

Reviewer 3 Report

-Uppercase the first letter of the first word in:

          - Titles: 4.4. fuel cells; 4.5. biosensors; ....;

          - Table 1. (Purpose); Table 3-4 (Sample);

          - Legends of the Figures;

- Improve the quality of the Figures 5-7-8;

- Letters are missing for the word "carbons" in the Figures 5-7;

- Align Tables 7-8-9 to the center of the page.

Author Response

Q: -Uppercase the first letter of the first word in:

          - Titles: 4.4. fuel cells; 4.5. biosensors; ....;

          - Table 1. (Purpose); Table 3-4 (Sample);

          - Legends of the Figures;

Answer: The first letters of the first words were uppercased for all the mentioned topics.

----------------------------------------------------------------------------- 

Q: - Improve the quality of the Figures 5-7-8;

Answer: The quality of the Figures were improved.

---------------------------------------------------------------

Q: - Letters are missing for the word "carbons" in the Figures 5-7;

Answer: Letters were added to Figures 5-7.

----------------------------------------------------------------

Q:- Align Tables 7-8-9 to the center of the page.

answer: Tables 7-8-9 were aligned to the center of the page.